

# Genetic models reveal historical patterns of sea lamprey population fluctuations within Lake Champlain

Cassidy C. D'Aloia[1,2], Christina B. Azodi[3,4], Sallie P. Sheldon[2], Stephen C. Trombulak[5] and William R. Ardren[6]

[1] Department of Ecology & Evolutionary Biology, University of Toronto, Toronto, Ontario, Canada
[2] Department of Biology, Middlebury College, Middlebury, VT, USA
[3] Department of Plant Biology, Michigan State University, East Lansing, MI, USA
[4] Department of Molecular Biology and Biochemistry, Middlebury College, Middlebury, VT, USA
[5] Department of Biology and Program in Environmental Studies, Middlebury College, Middlebury, VT, USA
[6] Western New England Complex, US Fish and Wildlife Service, Essex Junction, VT, USA

Corresponding author
Cassidy C. D'Aloia,
cassidy.daloia@gmail.com

## ABSTRACT

The origin of sea lamprey (*Petromyzon marinus*) in Lake Champlain has been heavily debated over the past decade. Given the lack of historical documentation, two competing hypotheses have emerged in the literature. First, it has been argued that the relatively recent population size increase and concomitant rise in wounding rates on prey populations are indicative of an invasive population that entered the lake through the Champlain Canal. Second, recent genetic evidence suggests a post-glacial colonization at the end of the Pleistocene, approximately 11,000 years ago. One limitation to resolving the origin of sea lamprey in Lake Champlain is a lack of historical and current measures of population size. In this study, the issue of population size was explicitly addressed using nuclear (nDNA) and mitochondrial DNA (mtDNA) markers to estimate historical demography with genetic models. Haplotype network analysis, mismatch analysis, and summary statistics based on mtDNA noncoding sequences for NCI (479 bp) and NCII (173 bp) all indicate a recent population expansion. Coalescent models based on mtDNA and nDNA identified two potential demographic events: a population decline followed by a very recent population expansion. The decline in effective population size may correlate with land-use and fishing pressure changes post-European settlement, while the recent expansion may be associated with the implementation of the salmonid stocking program in the 1970s. These results are most consistent with the hypothesis that sea lamprey are native to Lake Champlain; however, the credibility intervals around parameter estimates demonstrate that there is uncertainty regarding the magnitude and timing of past demographic events.

## INTRODUCTION

The origin of the landlocked population of sea lamprey (*Petromyzon marinus*) in Lake Champlain has been the subject of an ongoing debate in recent years (*Bryan et al., 2005*; *Waldman, Grunwald & Wirgin, 2006*; *Waldman et al., 2009*; *Eshenroder, 2009*; *Eshenroder, 2014*). The sea lamprey is an anadromous fish that has a parasitic juvenile phase during which it feeds on the bodily fluids of a variety of prey fishes, including large salmonids such as lake trout (*Salvelinus namaycush*), Atlantic salmon (*Salmo salar*), and lake whitefish (*Coregonus clupeaformis*), as well as lake sturgeon (*Acipenser fulvescens*). Recent research has suggested that native coastal populations of sea lamprey may have positive environmental impacts on freshwater streams. For example, post-spawning sea lamprey carcasses may be important sources of marine-derived nutrients and materials in oligotrophic streams (*Guyette et al., 2014*). However, the overall effect of landlocked sea lamprey populations in the Great Lakes and Lake Champlain has been detrimental. Lamprey-induced collapses of native fish populations have been well documented in the region since the 1970s (*Smith, 1971*; *Smith & Tibbles, 1980*).

Consequently, control efforts were developed in the Great Lakes to suppress sea lamprey populations and facilitate restoration of native species. Control methods are varied and include widespread biocide use, physical migration barriers, and spawning-phase traps in tributaries. Although these methods have been criticized for the potential negative effects that non-target species may experience (*McLaughlin, Marsden & Hayes, 2003*), the effort has been largely successful in terms of lamprey control. Using the methods developed in the Great Lakes, an experimental lamprey control program was implemented in Lake Champlain in 1990, followed by a long-term program beginning in 2001. Although the control methods in Lake Champlain are similar to those in the Great Lakes, a major difference is the consensus regarding the fish's status as native versus invasive; in the Great Lakes, sea lamprey are known to be invasive (with the exception of Lake Ontario), while much debate surrounds the population in Lake Champlain.

To date, two alternative historical scenarios have dominated the discussion regarding the origin of sea lamprey in Lake Champlain. First, it has been argued that individuals from the Atlantic coast population invaded Lake Champlain via the canal system sometime between the 1840s and the 1920s (*Eshenroder, 2009*; *Eshenroder, 2014*). This hypothesis is based on the fact that the first documentation of the species in the lake was in 1929 (*Eshenroder, 2014*), 13 years after the construction of the Champlain Barge Canal (the third version of the Champlain Canal), which directly connected Lake Champlain to the Hudson River. Second, recent genetic analyses based on both nuclear (nDNA) and mitochondrial DNA (mtDNA) markers suggest that sea lamprey are native to Lake Champlain (*Bryan et al., 2005*; *Waldman, Grunwald & Wirgin, 2006*; *Waldman et al., 2009*). *Bryan et al. (2005)* tested alternative coalescent-based colonization models using microsatellite markers and found evidence of long-term vicariance in the Lake Champlain population, concluding that the most probable route of entry was via the St. Lawrence River upon the initial formation of Lake Champlain about 12,500 years ago. *Waldman, Grunwald & Wirgin (2006)* compared haplotype frequencies of the Lake Champlain population to Atlantic Coast and Great

Lakes populations using mitochondrial non-coding DNA and concluded that the data were most consistent with a post-glacial colonization during a period when modern-day Lake Champlain was an arm of the Atlantic Ocean called the Champlain Sea. However, neither hypothesis has been widely accepted throughout the scientific and management communities. The "nonnative" hypothesis has been criticized because it is based on the absence of species documentation data during the 1800s—a time when systematic biological censuses were not conducted in the lake—and because it lacks due consideration of the post-glacial geological history of the region. Likewise, the "native" hypothesis has recently been called into question after an extensive review of historical documentation suggested that the time of origin used in genetic models (1841) was based on an erroneous species identification and, subsequently, may have biased the results (*Eshenroder, 2014*).

As a result of this ongoing debate, prior research has focused exclusively on the timing of the origin of *P. marinus* in Lake Champlain, while little is known about the rest of the population's history. The two previous genetic studies of *P. marinus* in the lake were regional studies that compared the Lake Champlain sea lamprey population to other populations from the Great Lakes and the Atlantic Ocean (*Bryan et al., 2005*; *Waldman, Grunwald & Wirgin, 2006*). While both studies found evidence for long-term vicariance from anadramous and other freshwater populations, it has proven challenging to reconcile the results with the complete lack of historical documentation of sea lamprey in Lake Champlain (*Bryan et al., 2005*). Given that these regional studies have already shown the Lake Champlain population to be differentiated from all other populations, additional analyses can further probe the genetic data by conducting rigorous intra-population analyses to model historical population dynamics *within* Lake Champlain. An objective assessment of the timing and magnitude of fluctuations in population size over time may therefore provide a more complete understanding of the history of *P. marinus* in Lake Champlain, while simultaneously shedding new light on the contentious topic of whether or not the species is invasive.

In cases such as this one, where historical census data are unavailable, genetic markers can be powerful tools for inferring demographic fluctuations. These inferences are possible because the census population size ($N_c$) is generally proportional to the effective population size ($N_e$). Effective population size is the size of an idealized population (i.e., with binomial variance in reproductive success, an equal sex ratio, and discrete generations), that is subject to the same level of genetic drift and inbreeding as the census population (*Wright, 1938*). Drawing on the relationship between effective and census population size, and the fact that rapid demographic fluctuations can be detected with genetic markers, we can investigate how effective population size has changed over time and, in turn, infer proportional changes in the overall population (*Waples, 1989*; *Frankham, 1995*).

Traditionally, genetic approaches to estimating historical demography have used summary statistics to test whether extant population-level data deviate from theoretical expectations under alternative models of population stasis, contractions, and expansions (*Cornuet & Luikart, 1996*; *Harpending et al., 1998*; *Schneider & Excoffier, 1999*; *Garza & Williamson, 2001*. For example, mismatch analysis uses sequence data to compare

the distribution of observed pairwise differences between all haplotypes in a population to the distribution expected under a specified population change. Expansions, contractions, and equilibrium each generate a particular pattern of the distribution of pairwise differences among sequences. Contractions or equilibrium lead to multimodal, ragged distributions while expansions result in a smooth unimodal Poisson distribution of pairwise differences (*Harpending et al., 1998*; *Schneider & Excoffier, 1999*). These moment-based metrics are widely used because they are easy to obtain with sequence and/or allele frequency data. However, they provide crude approximations of population changes, and their precision is linked to the timing and magnitude of the demographic change in question.

Arguably, a more powerful approach to inferring past demographic change is coalescent modeling (*Storz & Beaumont, 2002*; *Beaumont & Rannala, 2004*). Coalescent theory seeks to describe the ancestral relationship of a particular gene or set of genes by recognizing that the probability of two lineages coalescing during a particular generation is inversely proportional to effective population size at that time (*Beaumont & Rannala, 2004*; *Kuhner, 2008*). Thus, these models trace separate genetic lineages back to their most recent common ancestor (*Kuhner, 2008*) and connect these genealogies to changes in effective population size (*Storz & Beaumont, 2002*). Models such as BEAST and Msvar adopt Bayesian Markov chain Monte Carlo (MCMC) methods to explore parameter space and sample the posterior distributions of the demographic parameters of interest (*Beaumont, 1999*; *Drummond & Rambaut, 2007*). A key advantage to these methods is the ability to include time as one of the estimated parameters, as opposed to using fixed time points as assumptions in the model. Directly estimating time is particularly important for models of historical sea lamprey demography, as previous genetic studies that focused on the population's origin have been criticized for using fixed, potentially-incorrect dates (*Eshenroder, 2014*).

The sea lamprey is a tractable study species for coalescent modeling of effective population size fluctuations because several genetic resources are available. The entire mitochondrial genome is sequenced (*Lee & Kocher, 1995*), enabling researchers to sequence hypervariable regions of mitochondrial DNA (mtDNA). For decades, mtDNA has been widely used to infer demographic processes because of its maternal inheritance, small effective population size, and relatively fast rate of evolution (*Avise, 1994*; *White et al., 2008*). These characteristics are particularly useful for demographic studies because uniparental inheritance can be modeled without the complications of recombination, and signatures of relatively recent demographic events are more readily detectable when effective population size is small and mutation rate is elevated. Also, *P. marinus* has two non-coding regions in the mitochondrial genome: non-coding region one (NCI) is 491-bp long and non-coding region two (NCII) is 199-bp long (*Lee & Kocher, 1995*). Finally, a suite of microsatellite markers has already been developed for this species (*Bryan et al., 2005*); thus, markers from the mitochondrial and nuclear genomes can be used concurrently to study the population's history. The concurrent use of two genomic marker types enables intra-population replication for inferring demographic history (*Eytan & Hellberg, 2010*).

The purpose of this study was to use multiple analytical approaches and two sets of genetic markers to investigate historical population fluctuations in Lake Champlain sea
lamprey. First, moment-based methods—including a mismatch distribution and Fu's $F_s$ statistic—were used to generate coarse estimates of historical population expansions and/or contractions based on mtDNA sequence data (NCI and NCII). Second, two coalescent MCMC models were used to explicitly estimate changes in effective population size over time. Mitochondrial sequence data (NCII) were used to generate a Bayesian Skyline Plot (BSP) in the program BEAST to model effective population size history while taking into account coalescent and phylogenetic uncertainty. Previously-published allele frequency data for eight nuclear microsatellite loci (*Bryan et al., 2005*) were also used in the program Msvar to estimate four demographic parameters: historical effective population size, current effective population size, mutation rate, and time. In total, we used 10 loci ($n = 2$ non-coding mtdna; $n = 8$ nuclear microsatellites), and employed both moment-based methods as well as two classes of coalescent models to explore historical demography. Taken together, these data can provide insight into signatures of demographic events within two separate genomes.

## METHODS

### Sample collection and mitochondrial DNA sequencing

To estimate historical population fluctuations within Lake Champlain, fin tissue samples from spawning-phase sea lamprey were obtained from the US Fish and Wildlife Service (USFWS) in May–June 2009. All tissue collection was conducted as part of routine USFWS sampling; the authors of this study received tissue, but did not handle any vertebrate specimens. First, to estimate fluctuations based on mtDNA sequence data, samples were collected at three Lake Champlain tributaries: Great Chazy River ($n = 33$), Malletts Creek ($n = 33$), and Beaver Brook ($n = 28$), representing the northern, central, and southern regions of Lake Champlain, respectively. A broad geographic sampling regime was used in order to test for population structure. All tissue samples were stored in 95% non-denatured ethanol and genomic DNA was extracted using DNeasy Blood and Tissue Kits (Qiagen, Venlo, Netherlands).

The two non-coding regions of the mtDNA genome, NCI and NCII, were amplified in all Lake Champlain samples using PCR with the lamprey-specific primers CR1 (*Waldman et al., 2004*) and LampR (5′-AATAGACGGTTGGTGGGACA-3′). PCR reactions were performed in 25 μl volumes with the following reagents: 0.2 μl Qiagen *taq* polymerase (5 units/μl), 2.5 μl 10X PCR buffer (with 1.5 mM MgCl$_2$), 10 μM each primer, 10 μM dNTP, and 50–100 ng template DNA. Thermal cycler settings were set at an initial denaturation at 95 °C for 5 min followed by 40 cycles of 95 °C for 45 s, 56.5 °C for 45 s, and 72 °C for 1 min; and a final extension at 72 °C for 10 min. PCR products were cleaned using the Wizard SV Gel and PCR Clean-Up Kit (Promega, Madison, Wisconsin, USA) and eluted in 25 μl H$_2$O.

To sequence the Lake Champlain samples at both non-coding regions, forward extension reactions were run for each individual using the forward primers CR1 (for NCI) and CASSFN (for NCII) (5′- GACCCCTAAGTTCATTGC-3′). All primers designed specifically for this study, including LampR, CR1, and CASSFN, were designed

using Primer3 (*Untergrasser et al., 2012*). Extension reactions were prepared using 1/4 reactions from the BigDye Terminator v3.1 Cycle Sequencing Kit (Applied Biosystems, Foster City, California, USA). For each 20 μl reaction, the following reagents were combined: 2 μl Ready Reaction Mix, 3 μl 5× sequencing buffer, 3.2 μl primer (1 μM), 1 μl DNA template (20–30 ng), and 10.8 μl water. Thermal cycler settings were set at an initial denaturation at 96 °C for 1 min followed by 30 cycles of 96 °C for 20 s, 50 °C for 20 s, and 60 °C for 4 min. Reactions were cleaned with 2 μl of 2.2% sodium dodecyl sulfate and returned to the thermal cycler at 98 °C for 5 min followed by 25 °C for 10 min. Reactions were then purified in Sephadex size-exclusion columns (GE Healthcare Life Sciences, Little Chalfont, UK) and 12 μl of purified products were loaded into an Applied Biosystems 3130 Genetic Analyzer for sequencing.

Sequences from individuals that successfully amplified at both non-coding regions ($n = 54$) were concatenated into a single sequence denoted by $NC_{total}$. Concatenated sequences were then trimmed to a 652-bp region to exclude repetitive portions of reads prone to slippage (NCI: excluded 14 bp before position 15382; NCII: excluded 28 bp after position 16173). Regions prone to slippage were identified by superimposed sequences on the chromatograms. All trimmed sequences were aligned with the MUSCLE algorithm (*Edgar, 2004*). Finally, a haplotype network was generated for all $NC_{total}$ haplotypes to visualize their relationship using TCS v.1.18 (*Clement, Posada & Crandall, 2000*).

Second, to estimate population fluctuations based on nuclear microsatellite data, we used previously-published allele frequency data (*Bryan et al., 2005*). *Bryan et al. (2005)* genotyped individuals from Great Chazy River ($n = 40$) and Lewis Creek ($n = 40$) at 8 microsatellite loci. There was no evidence for linkage disequilibrium or significant deviations from Hardy–Weinberg equilibrium (see *Bryan et al., 2005* for more details).

## Moment-based analyses of historical population changes

Historical population fluctuations were inferred using three moment-based methods. First, the mismatch distribution of pairwise differences between $NC_{total}$ haplotypes was plotted. The observed distribution of pairwise differences was compared to the expected number of pairwise differences under a model of population expansion with 1,000 bootstrap replicates in Arlequin v.3.1 (*Excoffier, Laval & Schneider, 2005*). Second, Harpending's raggedness index ($r$) was used to test whether the observed distribution was significantly different from the expected theoretical distribution under a model of expansion (*Harpending, 1994*). Third, an alternative metric, Fu's $F_s$ statistic, was used to test the selective neutrality of mutations (*Fu, 1997*). Fu's $F_s$ can detect an excess (or deficiency) of haplotypes, given the observed haplotype diversity, thereby indicating a population expansion (or contraction).

## Coalescent analyses of historical population changes

Two Bayesian coalescent MCMC models were used to estimate historical demographic fluctuations of *P. marinus* in Lake Champlain over time. First, the program BEAST v.1.6.2 (*Drummond & Rambaut, 2007*) was used to make inferences based on NCII sequences. This region was selected because of its high concentration of polymorphic sites. BEAST

applies a Bayesian coalescent-based procedure, using MCMC to sample the posterior distribution of genealogical trees, demographic parameters over time, and coalescent events given sequence information and a set of priors. For our demographic model, we applied the Bayesian Skyline Plot (BSP). BSP is a change-point model that, assuming a single panmictic population, estimates fluxes in population size through time and uses a smoothing procedure to visualize these changes (*Drummond et al., 2005*). To determine which nucleotide substitution model fit the data, Akaike information criteria (AIC) values were calculated in jMODELTEST v.0.1.1 (*Guindon & Gascuel, 2003*). The substitution model selected by AIC was then used as a prior in BEAST. For our baseline study, we assumed a strict molecular clock of $3.6 \times 10^{-8}$ substitutions per base per year (i.e., 3.6% substitutions per million years), based on previous estimates of divergence in mtDNA noncoding regions in fishes (*Donaldson & Wilson, 1999*). The BSP group number ($m$) was set to 15. The parameter $m$ allows adjacent coalescent intervals to be grouped so that they can have the same $N_e$—it serves to smooth the resulting BSP. The maximum effective population size was set at 10,000, a high estimate based on preliminary coalescent model runs. MCMC chains were run for $50^6$ iterations in triplicate, sampling the posterior distribution every 1,000 iterations. The sampling distribution of the model was evaluated in TRACER v.1.5, with the first 10% discarded as burn-in (*Rambaut et al., 2014*). Quality of the MCMC convergence was assessed by the effective sample sizes (ESS): if the ESS value was less than 100, it was assumed that the MCMC chain had not been run long enough to get an accurate representation of the posterior distribution and the trace was discarded (*Drummond et al., 2007*).

A sensitivity analysis for the BEAST modeling was carried out on two parameters— clock rate and maximum effective population size—to assess whether priors biased parameter estimates. These parameters were selected because the priors were based on our preliminary estimates. For each parameter change, three additional MCMC chains were run keeping all other baseline input values constant. We used two alternative maximum population sizes (20,000 and 100,000) and one alternate clock rate ($2.0 \times 10^{-8}$ substitutions per base per year, i.e., 2.0% substitutions per million years). To determine how sensitive model results were to the priors, runs with altered priors were compared.

Historical demographic changes were also inferred from coalescent modeling of nuclear microsatellite data. Allele frequencies at eight microsatellite loci, previously published by *Bryan et al. (2005)*, were used as input for the Bayesian MCMC model Msvar v.1.3 (*Beaumont, 1999*). Msvar uses probable genealogies of allele frequency data to generate posterior probability distributions of four demographic parameters: current effective population size ($N_0$), historical effective population size ($N_1$), mutation rate ($\mu$), and time since the demographic change began ($t$) (*Storz & Beaumont, 2002*; *Beaumont & Rannala, 2004*). Broad prior distributions were defined for each parameter to test whether the model could detect true population fluctuations (Table S1). Five independent chains were run for a panmictic sea lamprey population (*Bryan et al., 2005*; *Waldman, Grunwald & Wirgin, 2006*) under a model of exponential growth, with an average generation time of six years (*Hardisty & Potter, 1971*), and a $N_e/N_c$ ratio of 0.2 (*Frankham, 1995*). Each chain consisted

**Table 1 Polymorphic sites among 14 mtDNA haplotypes from the concatenated non-coding region sequences (NC_total), relative to the most common haplotype (first entry).** Base pair positions relative to the reference mitochondrial genome are provided (*Lee & Kocher, 1995*).

| | | NCI | | | NCII | | | | | | | | |
|---|---|---|---|---|---|---|---|---|---|---|---|---|---|
| ID | Genbank accession # | 15398 | 15402–15403 | 15403 | 16049–16050 | 16062 | 16112 | 16114 | 16115 | 16134 | 16139 | 16142 | 16169 |
| 1 | GU459340 | G | – | C | T | T | T | T | C | T | T | C | C |
| 2 | GU459341 | . | C | . | . | . | . | . | . | . | . | . | . |
| 3 | GU459342 | . | – | . | . | – | . | . | . | . | . | . | . |
| 4 | GU459343 | . | – | . | . | . | . | . | T | . | . | T | T |
| 5 | GU459344 | . | – | . | . | . | A | . | – | . | . | T | T |
| 6 | GU459345 | . | – | . | – | . | . | . | . | . | . | . | . |
| 7 | GU459346 | . | – | . | . | . | . | . | . | . | . | . | T |
| 8 | GU459347 | . | – | . | . | . | . | . | . | A | . | – | T |
| 9 | GU459348 | A | – | . | . | . | . | . | . | . | . | . | . |
| 10 | GU459349 | . | – | . | . | . | . | . | . | . | . | – | T |
| 11 | GU459350 | . | – | – | – | . | . | . | . | . | . | . | . |
| 12 | GU459351 | . | – | . | . | . | . | . | – | . | . | T | T |
| 13 | GU459352 | . | – | . | . | . | . | – | – | . | A | T | T |
| 14 | GU459353 | . | – | . | . | . | . | . | T | . | A | T | . |

**Notes.**

Dots indicate no change, dashes indicate a deletion, and G/C/A/T represent point mutations.

of $8 \times 10^8$ iterations, with sample points taken once every 10,000 iterations. The burn-in portion of the chain was excluded by truncating runs to consider only the second half. To estimate each demographic parameter, the five truncated chains were combined into one posterior distribution and peak density values were recorded for each parameter, along with 95% highest probability density (HPD) credibility intervals. A Bayes factor was also calculated to test whether there was more support for a population expansion or contraction using the method described by *Storz & Beaumont (2002)*. Under the Bayesian statistical framework, a Bayes factor is an appropriate metric for comparing alternative models.

Sensitivity analyses were not conducted for Msvar because the model is computationally demanding; however, simulation modeling has revealed that Msvar has the power to detect true population contractions even with broad priors (*Girod et al., 2011*). These simulation models have also shown that the precision of estimates increases when demographic parameters are scaled using coalescent theory. Thus the marginal posterior distributions of the following scaled parameters were also plotted: $\theta_0 = 4N_0\mu$ and $\theta_1 = 4N_1\mu$ (effective population sizes scaled by mutation rate); $t_f = t/(2N_0)$ (time scaled by current effective population size).

## RESULTS

### Haplotype diversity & population structure

We observed 14 unique haplotypes among the 54 NC_Total mitochondrial sequences (Table 1). The haplotype network of all 14 NC_total haplotypes visualizes their relationships as well as their relative abundance (Fig. 1). Haplotype 1 is predominant in Lake Champlain,
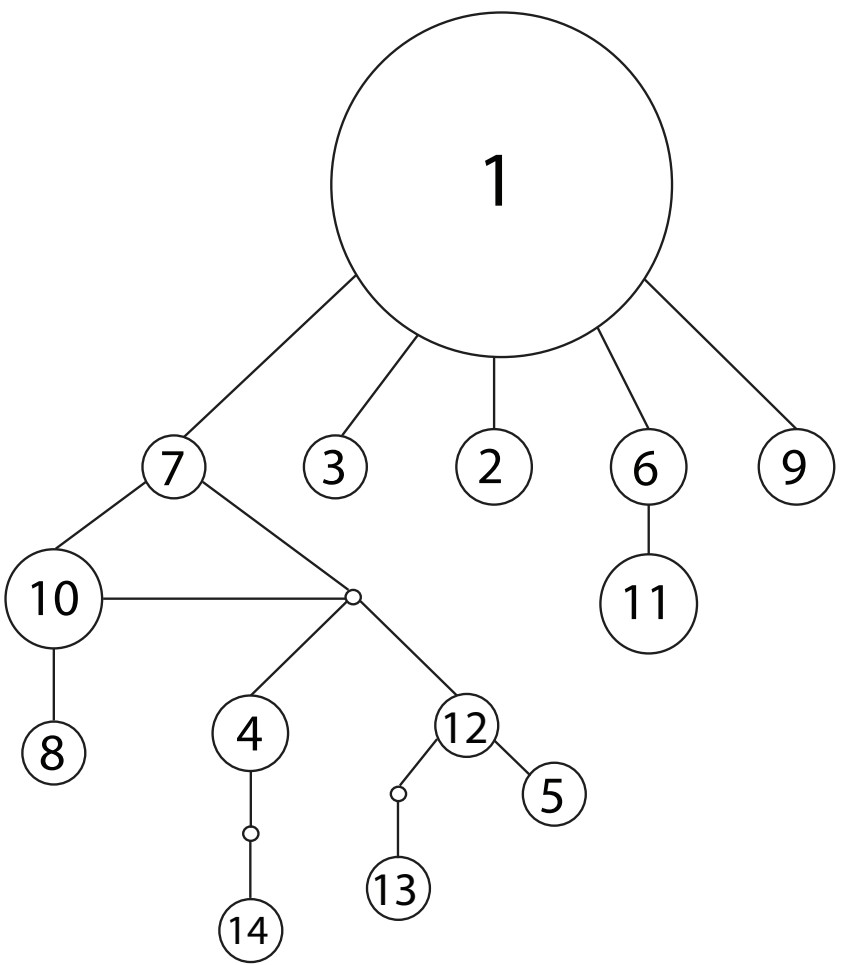

**Figure 1 Haplotype network for 14 concatenated mtDNA haplotypes found in Lake Champlain, constructed using TCS v1.2.1 with 95% parsimony.** The size of the circle is proportional to the relative abundance of the haplotype. Solid lines represent one point mutation and small, unfilled circles represent inferred haplotypes. The mutations resulting in branching off of haplotype 7 are concentrated in the repetitive region of the NCII 3′ region, which has an elevated mutation rate due to strand slippage.

with five other rare haplotypes (2, 3, 6, 7, and 9) having only one base pair difference from haplotype 1. Most of the mutations within these five haplotypes are point mutations within NCI or the 5′ end of NCII (Table 1). In contrast, most of the remaining rare haplotypes branch off of haplotype 7 and are characterized by mutations within the A/T-rich, repetitive 3′ region of NCII. In general, these mutations in the repetitive region were retained in each subsequent repeat of the sequence, leading to a larger number of overall mutations in NCII.

An exact test of population differentiation revealed that haplotypes were randomly distributed across the three sampling locations ($n = 54$; $p = 0.265$; # dememorization steps $= 10,000$; # steps Markov chain $= 100,000$). Thus, the Lake Champlain population is considered to be panmictic, consistent with results from previous studies (*Bryan et al., 2005*; *Waldman, Grunwald & Wirgin, 2006*).

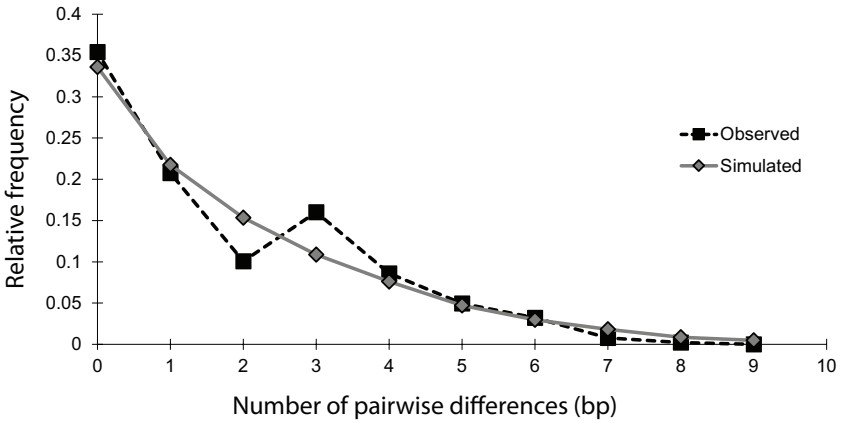

**Figure 2 Mismatch distribution of the 14 concatenated mtDNA haplotypes, conducted in Arlequin v.3.1.** The black squares represent the observed number of pairwise bp differences between haplotypes, while the gray diamonds represent the expected number of pairwise bp differences between haplotypes, based on a model of population expansion (bootstrap replicates = 1,000). The relatively smooth and unimodal shape of the observed distribution closely matches the expected distribution for a demographic expansion.

## Moment-based analyses of historical population changes

The mismatch distribution of $NC_{total}$ was smooth and unimodal, suggesting that the sea lamprey population has undergone an expansion (Fig. 2). The distribution of observed pairwise differences closely matches the expected distribution of pairwise differences under a model of population expansion, with a raggedness index that was positive, but not significant ($r = 0.04$; $n = 54$; $p = 0.86$). This indicates that there was no significant deviation from the theoretical model of expansion. Additionally, Fu's $F_s$ statistic was significantly negative ($F_s = -6.61$; $n = 54$; $p = 0.02$), indicating an excess of rare haplotypes, which would be predicted under a scenario of a recent population expansion. Overall, these moment-based methods of studying historical demography with mtDNA data strongly support a population expansion, without explicit estimates of the timing and magnitude of the event.

## Coalescent analyses of historical population changes

For the BEAST analysis of mtDNA NCII data, the best-fit model of sequence evolution determined by jMODELTEST was Hasegawa, Kishino and Yano (HKY) + I, where I means there is a significant proportion of invariable sites. The BSP derived from the NCII data shows a decrease in effective population size starting around 400 years ago that continued until 50 to 100 years ago, at which point effective population size slowly began to increase (Fig. 3). The sensitivity analysis showed that the maximum population size prior was directly proportional to the current $N_e$ estimate; when doubled, the $N_e$ estimates also doubled. However, changes in the maximum population size and clock rate priors did not affect the overall pattern of the BSP (Table S2).

In contrast to the BEAST analysis, coalescent modeling of microsatellite alleles estimated one historical population contraction. The five Msvar chains converged below a critical

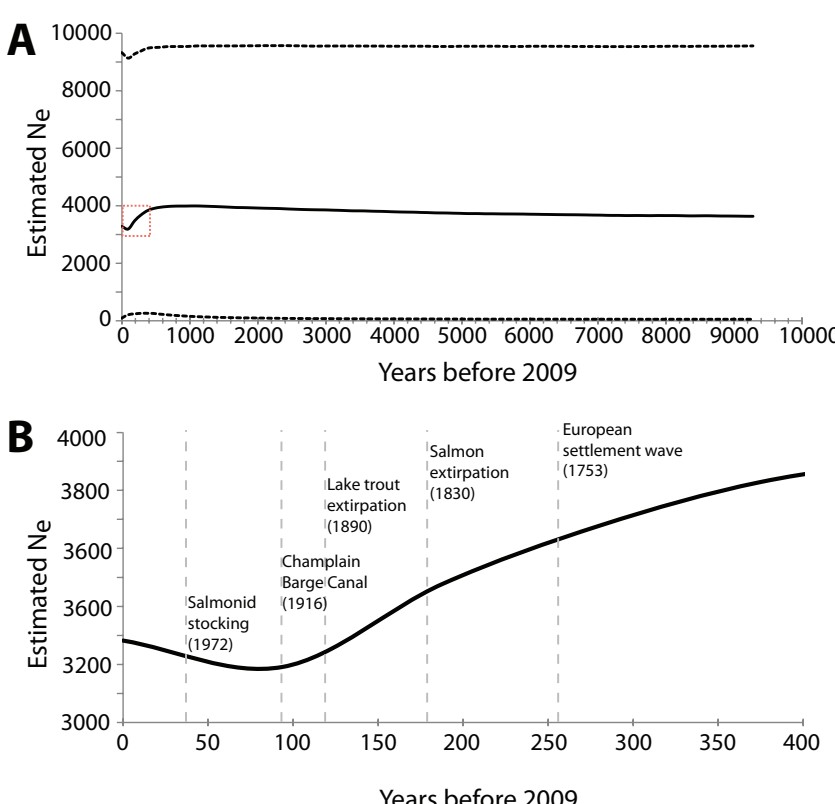

**Figure 3 Bayesian skyline plot (BSP) derived from NCII sequence alignments from Lake Champlain lamprey collected in May–June 2009.** (A) The mean effective population size is estimated from the Bayesian posterior distribution and is shown as the thick solid line. The horizontal dashed lines show the 95% HPD intervals around the $N_e$ estimate. The dashed red box highlights the portion of the BSP shown in B; (B) Zoom-in of the BSP for 400 years prior to 2009. The dashed vertical gray lines represent the timing of relevant historical events in the region. Note that the $x$ and $y$ axes differ between the two panels.

potential scale reduction factor of 1.2 (*Brooks & Gelman, 1998*) for all four demographic parameters, indicating that there was good convergence of parameter estimates. The Bayes factor (BF) for a population contraction indicated very strong support for a population contraction over a population expansion (2*ln(BF) = 10.3) (*Kass & Raftery, 1995*), with the vast majority of MCMC iterations estimating a population contraction (Fig. 4A). The posterior density distributions reveal the estimates for each demographic parameter with 95% HPD intervals (Figs. 4B–4D). There was a decrease in effective population size from approximately 2,660 (HPD$_{95\%}$ = 153, 65750) down to approximately 50 currently (HPD$_{95\%}$ = 0.06, 1440) (Fig. 4B). The density peak was higher, with narrower credibility intervals, for historical effective population size compared to current effective population size. This population contraction was estimated to have begun approximately 820 years ago, although the peak density was only 40% for this time parameter, with wide credibility intervals (HPD$_{95\%}$ = 0.81, 71558) (Fig. 4C). Finally, there was a high density peak for the mutation rate estimate, with over 80% of the estimates approaching a modal value of $3.26 \times 10^{-4}$ mutations/site/generation (HPD$_{95\%}$ = $3.52 \times 10^{-5}$, $2.78 \times 10^{-3}$) (Fig. 4D).

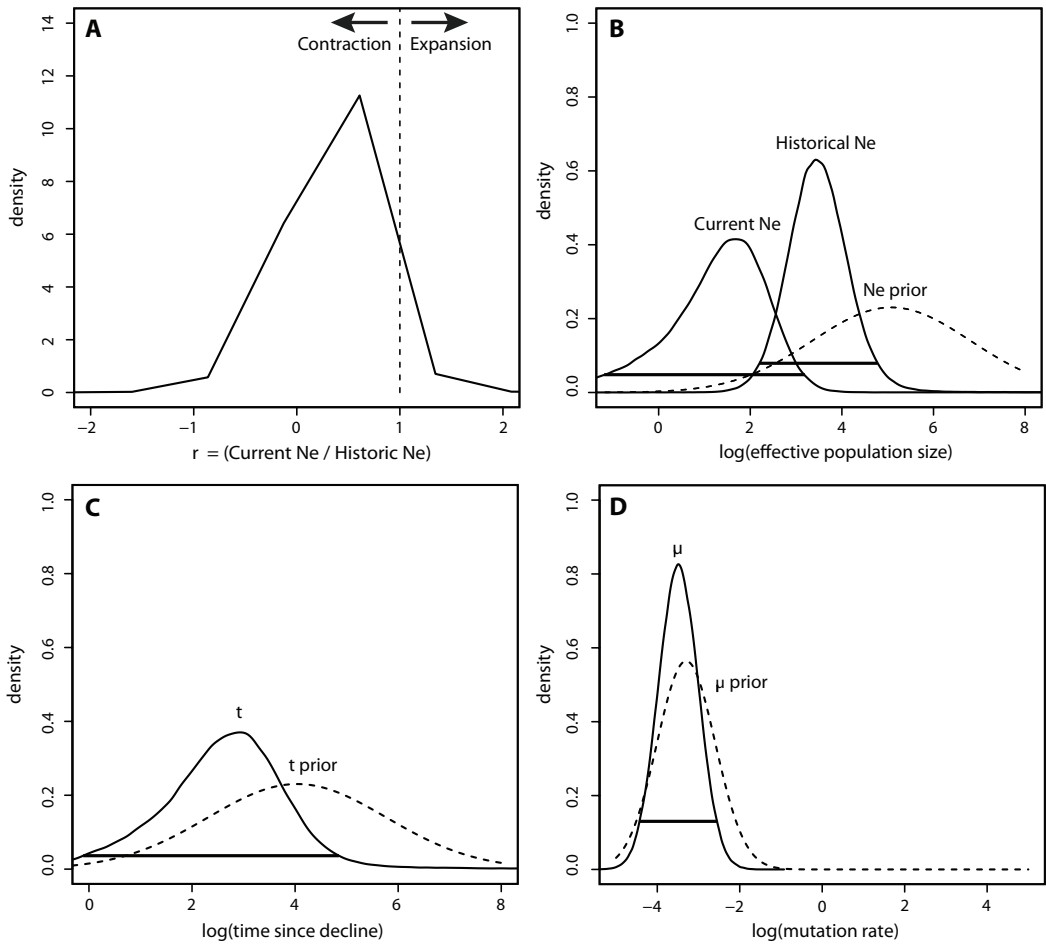

**Figure 4 Msvar results based on microsatellites.** (A) Density plot across MCMC iterations for $r$, which represents the ratio of current $N_e$ to historical $N_e$, with $r > 1$ indicating a population expansion and $r < 1$ indicating a population contraction. (B–D) Posterior density distributions for each demographic parameter, plotted on a $\log_{10}$ scale. Solid lines represent posterior density distributions and dashed lines represent prior distributions (with solid horizontal lines representing the 95% HPD credibility intervals for the posterior distributions); (B) Effective population sizes; (C) time in years since population contraction; (D) mutation rate as # mutations/site/generation.

The posterior distributions of the scaled demographic parameters also indicate a population contraction (Fig. 5). As predicted by simulation analyses, scaling the parameters increased the precision of parameter estimates. The magnitude of the scaled population size estimates were very similar to the unscaled estimates. Modal $\theta$ estimates corresponded with a decline from approximately 2,335 individuals historically ($\text{HPD}_{95\%} = 359, 29614$) to 60 individuals currently ($\text{HPD}_{95\%} = 0.08, 772$). Despite increased precision, there was still some overlap in the 95% credibility intervals for $\theta$ (Fig. 5A). In contrast, there was a substantial increase in precision of the time estimate, suggesting a more ancient demographic event: the modal value of the scaled time distribution corresponded with an event beginning approximately 1,230 years ago ($\text{HPD}_{95\%} = 128, 7522$) (Fig. 5B).

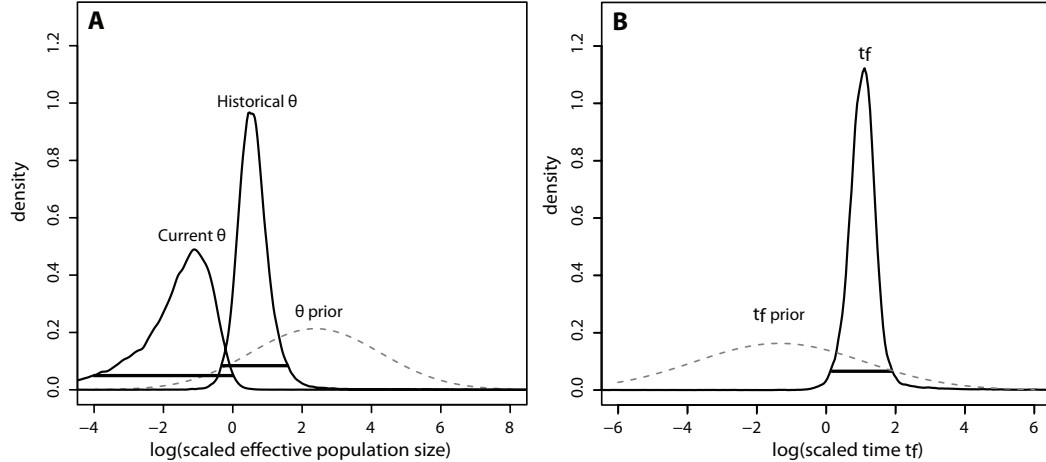

**Figure 5 Marginal posterior distributions of scaled parameters from Msvar.** Solid lines represent posterior density distributions and dashed lines represent prior distributions (with solid horizontal lines representing the 95% HPD credibility intervals for the scaled posterior distributions). (A) Scaled effective population sizes on a $\log_{10}$ scale; (B) scaled time on a $\log_{10}$ scale.

## DISCUSSION

Genetic-based methods can be useful tools for studying demographic changes in the absence of historical population records, though their precision is linked to the timing and magnitude of the events themselves (*Beaumont & Rannala, 2004*; *Lawton-Rauh, 2008*). In this study, we applied multiple analytical approaches and used two sets of genetic markers to investigate the population history of *P. marinus* in Lake Champlain. Synthesizing the results to draw conclusions about the most likely historic scenario of demographic change requires a critical analysis of the results generated by each method, an exploration of sea lamprey ecology, and consideration of the history of land-use and management practices in the region.

### Moment-based analyses of historical population changes

Mismatch distribution analysis is based on the assumption that demographic changes leave corresponding genetic signatures in neutral sequence data and gene trees (*Rogers & Harpending, 1992*; *Harpending et al., 1998*). The mismatch distribution of this study fits a model of population expansion well, being both smooth and unimodal. However, a major weakness of this method is its poor ability to make specific inferences about the timing of demographic changes. In general, when using moment-based methods, only a general inference as to whether the change was relatively recent or ancient can be made. The presence of many low-frequency mutations is one indication of a 'recent' expansion (*Schneider & Excoffier, 1999*). This pattern was evident in the haplotype network, in which the majority of haplotypes in Lake Champlain were present in only one to three individuals (Fig. 1). Additionally, the mean number of pairwise differences in the mismatch distribution can be used for a crude estimate of timing: a 'low' mean can indicate a 'recent' expansion while a 'high' mean can indicate a more 'ancient' expansion (*Okello et al., 2005*). The haplotypes in the Lake Champlain lamprey population had a mean of

1.74 mismatches, which supports a 'recent' expansion. Yet, without explicit guidelines for differentiating between 'low' and 'high,' or inferring what 'recent' and 'ancient' indicate on an evolutionary timescale, caution should be used in interpreting these results. Instead, these results are more appropriately used as supplemental evidence to be considered alongside inferences made by coalescent-based analyses.

## Coalescent analyses of historical population changes

BEAST and Msvar analyses both detected an initial decline in effective population size. These congruent results between two separate genomes strengthen the evidence for a decline; however, the magnitude and timing of the event(s) remain uncertain. The BSP generated using BEAST provides credibility intervals at every time point that reflect both coalescent and phylogenetic uncertainty. This proved to be important, as the BSP shows that the credibility intervals for effective population size ranged from 51 to 9,500. These wide credibility intervals around $N_e$ could be due to the limited number of segregating sites within the NCII sequence data. The sensitivity analysis also showed that the maximum population size prior affected the BSP results, with the estimated $N_e$ being proportional to the prior. However, the overall pattern of demographic change was consistent regardless of the priors (see Table S2), indicating that the estimated mean $N_e$ always showed a contraction followed by an expansion.

In contrast to BEAST, Msvar estimates four parameters simultaneously and can only detect the single most likely demographic trend over time based on the strongest genetic signal. Bayesian models, such as Msvar, require prior distributions for each parameter (*Storz & Beaumont, 2002*; *Beaumont & Rannala, 2004*). Generating priors can be difficult when no accurate estimates of demographic parameters are available, as is the case for sea lamprey in Lake Champlain. Estimates of current sea lamprey population size are poor because census data are inferred from wounding rates on prey species. Moreover, no historical documentation of sea lamprey population size in Lake Champlain exists. In the absence of reliable data, broad and equivalent priors were used for current and historical effective population size (*Goossens et al., 2006*). Furthermore, through simulation analyses, *Girod et al. (2011)* showed that Msvar is powerful at detecting population fluctuations even without informed priors if the demographic event was sufficiently large and ancient. The contraction detected in this analysis indicates that effective population size has declined from roughly 2,660 to 50 (a low but plausible estimate given the aggressive population control program and the fish's high fecundity). It is important to note that these values represent best estimates based on the mode of the posterior density distributions, but that the 95% credibility intervals of historical and current population size overlap due to the uncertainty associated with the current effective population size parameter. Scaling the population size parameters by the mutation rate did increase precision, but there was still overlap in the credibility intervals. Thus, for a comprehensive population history, the Msvar results must be interpreted alongside results from the BEAST and moment-based analyses.

## Scenario 1: evidence for fluctuations within a native population

Capitalizing on multiple analytical approaches and two unique genetic data sets used in this study, it is possible to infer multiple demographic fluctuations. While the mismatch distribution detected a population expansion and coalescent modeling using Msvar detected a decline, the results may not be in conflict. Instead, they may be detecting the two different demographic changes that are both evident in the BSP generated by BEAST. By considering known events in the ecological history of Lake Champlain and the approximate time scale estimated by each method, we present the most parsimonious demographic history of a native sea lamprey population in Lake Champlain.

The decline in effective population size detected by both coalescent models may correlate with land-use changes and fishing pressures that began after the arrival of European settlers in the mid-18th century. The BSP from BEAST indicated that the decline began approximately 400 years ago. Given that there is uncertainty around this time estimate, one hypothesis is that a gradual decline in population size began around 1753 when a large number of European settlers arrived in the area. At this time, human impact on the landscape increased with the onset of large-scale agricultural and clear-cutting practices and, later, the construction of mills and dams that would have limited upstream lamprey spawning migrations (*Klyza & Trombulak, 1999*). Moreover, the acceleration of the decline shown in the BSP correlates with the extirpation of salmonids in Lake Champlain—the primary food source of parasitic-phase sea lamprey. The last documentation of native Atlantic salmon in the basin was 1830, with lake trout extirpation following shortly thereafter in the 1890s (*Fisheries Technical Committee, 2009*). Thus, it is plausible that the population contraction detected by both coalescent models can be explained by known ecological changes in the region, though some caution is warranted given that the scaled Msvar distributions suggest that the decline was more ancient (ca. 1,230 years ago).

The mismatch distribution analysis may be detecting a more recent population expansion associated with salmonid stocking in Lake Champlain. This recent expansion is supported by the upward trend in effective population detected in the BSP in the last 50 years. A stocking program began for both Atlantic salmon and lake trout in 1972 and now focuses on stocking the main part of the lake with yearlings (*Marsden et al., 2003*; *Fisheries Technical Committee, 2009*). The reintroduction of their primary food source could have allowed the sea lamprey population to expand.

While this demographic history is plausible given the ecological context of Lake Champlain over the last 300 years, it is important to consider whether the data represent two real population changes, or whether these changes are artifacts of the models. To address this issue, an important consideration is the difference in mutation rates between the nuclear and mitochondrial markers used in the study, as faster-evolving markers generally have a greater power to detect more recent events. While the mitochondrial genome tends to have a higher mutation rate than the nuclear genome due to the mutagenic properties of respiration by-products and the limited DNA repair mechanisms of the mitochondrial genome (*White et al., 2008*), nuclear microsatellites have an even higher

average mutation rate due to strand slippage. Indeed, Msvar estimated a fast mutation rate for the microsatellites used in the study, with the narrow posterior distribution peaking at $3.24 \times 10^{-4}$ mutations/site/generation. Nevertheless, Msvar, a model that uses microsatellite markers exclusively, has been shown to have a bias towards detecting ancient declines in effective population size (*Beaumont, 1999*). Most recently, this tendency has been rigorously confirmed through simulation studies (*Girod et al., 2011*). Therefore, even if there were signatures of two demographic fluctuations in the population's history, Msvar, which can only identify a single event, is predicted to detect the more ancient decline. As to whether or not the mtDNA sequence data show a true signature of expansion within the last 40 years, some insight can be gained from a comparison between BEAST model inputs and outputs. Even though an extremely strict (i.e., slow) molecular clock was used as an input in this analysis, the posterior distribution of effective population size still shows an upward trend beginning approximately 50 years ago. Posterior estimates that deviate from prior predictions generally indicate a strong genetic signal. Taken together, these lines of evidence suggest that the population fluctuations detected are likely to be real.

## Scenario 2: evidence for a founder event in the early 20th century

Differentiating between founder events resulting from an invasion versus bottlenecks within native populations can be challenging, as both events are predicted to lead to a decline in effective population size and a loss of genetic diversity (*Nei, Maruyama & Chakraborty, 1975*). Thus, an alternative interpretation of the data is that the population contraction detected in Msvar represents a bottleneck immediately following an invasion from the anadromous Atlantic Coast population sometime after the completion of the Champlain Barge Canal in 1916, while the expansion detected in the moment-based and BEAST analyses represents the population boom in the 1970s. While the timing of such an invasion falls within the bounds of the BEAST and unscaled Msvar credibility intervals, both coalescent models assume a single panmictic population. Therefore, if the present-day Lake Champlain population is significantly differentiated from the anadromous population, the coalescent models could not accurately estimate historical changes beginning in the differentiated population of origin. Indeed, *Bryan et al. (2005)* found evidence for significant genetic structure between Lake Champlain and anadromous sea lamprey populations, and long-term vicariance for the Lake Champlain population.

The mitochondrial haplotype network can provide additional insight into founder events: the random sampling associated with a *recent* founder event should lead to the loss of rare haplotypes. Therefore, a recent founder event is unlikely given the observed network, which features a number of rare haplotypes branching off of a predominant ancestral haplotype (Fig. 1). In sum, the majority of the evidence from the genetic models suggests that the population is native to Lake Champlain, but uncertainty remains and a founder event associated with an invasion in the early 1900s cannot be completely excluded given the genetic data that are presently available.

## CONCLUSIONS AND FUTURE DIRECTIONS

This study builds upon previous research by modeling historical sea lamprey population fluctuations in Lake Champlain. Considering both coalescent models and moment-based genetic approaches, we conclude that multiple demographic events are likely to have occurred over the past 300 years. Importantly, however, there is a large amount of uncertainty around these estimates. While we argue that the data largely align with prior genetic studies and are most consistent with the native hypothesis, the wide credibility intervals around our estimates cannot exclude an alternative interpretation that a founder event occurred in the early 20th century.

As such, we propose two potential lines of future research aimed at resolving the residency debate. First, expanded genomic sampling could provide more accurate estimates of historical population sizes within Lake Champlain, as well as the timing of divergence between the Lake Champlain and Atlantic populations. Second, given the uncertain results from neutral genetic data, it would be useful to investigate genes that may be under selection, e.g., those regulating Na/K-ATPase pumps in the gills. Population genetic analyses, gene expression analyses, and physiological saltwater challenges could be undertaken in tandem. Together, these lines of research may provide further clarity to the history of sea lamprey in Lake Champlain.

## ACKNOWLEDGEMENTS

The authors thank US Fish and Wildlife Service professionals in Essex Junction, VT for providing sea lamprey tissue. We also thank Vicenta Hudziak and Livingston Burgess for assistance in DNA sequencing and Jeremy Ward, Wayne Bouffard, Steve Smith, and Brad Young for providing feedback. We are grateful to John Waldman, Amy Russell, and one anonymous reviewer for helpful comments on the manuscript. The findings and conclusions in the article are those of the authors and do not necessarily represent the views of the USFWS.

### Funding

This project was funded by the Middlebury College Senior Work Fund and two student research grants from the Lake Champlain Research Consortium. The funders had no role in study design, data collection and analysis, decision to publish, or preparation of the manuscript.

### Grant Disclosures

The following grant information was disclosed by the authors:
Middlebury College Senior Work Fund.
Lake Champlain Research Consortium.

### Competing Interests

The authors declare there are no competing interests.

## Author Contributions

- Cassidy C. D'Aloia and Christina B. Azodi conceived and designed the experiments, performed the experiments, analyzed the data, wrote the paper, prepared figures and/or tables, reviewed drafts of the paper.
- Sallie P. Sheldon and Stephen C. Trombulak conceived and designed the experiments, contributed reagents/materials/analysis tools, reviewed drafts of the paper.
- William R. Ardren conceived and designed the experiments, analyzed the data, contributed reagents/materials/analysis tools, reviewed drafts of the paper.

## Animal Ethics

The following information was supplied relating to ethical approvals (i.e., approving body and any reference numbers):

The authors received tissue samples from the US Fish and Wildlife Service as part of routine annual lamprey collections. As such, the authors never handled vertebrate animals and this project did not require IACUC approval from Middlebury College.

## DNA Deposition

The following information was supplied regarding the deposition of DNA sequences:

GenBank accession numbers GU459340–GU459353.

## Data Availability

Microsatellite allele frequency data are previously published, as noted in the text. MCMC priors are provided as a supplemental table.

## Supplemental Information

Supplemental information for this article can be found online at http://dx.doi.org/10.7717/peerj.1369#supplemental-information.

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
