# Peer review of "Genetic models reveal historical patterns of sea lamprey population fluctuations within Lake Champlain"

_PeerJ, doi:10.7717/peerj.1369_

## Round 0.1 · original submission · Minor Revisions

Overall this was a very nice and well put together paper detailing the demographic history of sea lamprey in Lake Champlain. The authors have analyzed the data in the manuscript well, but I agree with reviewer 2 that the amount of segregating sites is quite low. Conclusions have not been overstated and the discussion reads well. Please make the changes requested by the reviewers and I look forward to receiving a revised copy of your manuscript.

Reviewer 1 ·

Basic reporting

Word Document p. 16, Can you report the 95% HPDs for the scaled MSVAR parameters, as you did in the previous paragraph? These parameters are at the heart of your questions regarding demographic history.

Figure 3, Is there a way to enlarge the BSP for just the most recent 1000 years, so you can plot the times associated with the hypotheses in the Discussion on it?

Experimental design

Given the genetic differences between Lake Champlain and Atlantic lamprey populations, you could estimate the timing of their divergence using IM or IMa. This might help evaluate the Founder Event hypothesis you discuss.

Validity of the findings

You are correct in stating that confidence intervals around time estimates are quite large, but I think you use the correct amount of caution to not overstate your conclusions.

Additional comments

The background and explanation of methods is appropriate - a reader not familiar with the demographic history of lampreys in Lake Champlain and\or Coalescent methods will understand this manuscript. Your interpretation of the data is sound, though the large 95% HPDs limit your conclusions. These Coalescent methods are a great tool to model population history - I enjoyed this MS.

·

Basic reporting

The manuscript was clearly written and appropriately organized. The authors did a good job of explaining the management issues surrounding the historical demography of Lake Champlain sea lampreys and previous work that has been done on this system. Figures were clearly designed, and all were necessary for the paper.

Experimental design

The authors clearly addressed the research question of the historical demography of Lake Champlain sea lampreys. The analyses were appropriate, both in scope and rigor. I had one minor question regarding the description of the BSP analyses:
• Line 264: I’ve run BSP analyses, and I don’t know what the authors mean by group number. Is this one of the operators? This phrasing needs to be clarified.

Validity of the findings

I was concerned about the low amount of variation in the mitochondrial data. Between the two regions, there are only 12 segregating sites, 5 of which are indels. Beast treats indels as missing data, and so the BSP analyses are based on only 7 variable sites. However, I think the analyses were conducted appropriately, and the authors adequately addressed the limitations of their data, both in the text and with the sensitivity analyses.

Additional comments

Minor comments:
• Line 235: equilibrium should not be capitalized.
• Line 330: delete space in 54.

---

## Round 0.2 · accepted · Accept

Thank you for making the requested changes to your manuscript. This was a high quality submission and a pleasure to read.